# The Relic *Trochodendron aralioides* Siebold & Zucc. (Trochodendraceae) in Taiwan: Ensemble Distribution Modeling and Climate Change Impacts

**Cheng-Tao Lin** [1] and **Ching-An Chiu** [2,*]

[1]  Department of Biological Resources, National Chiayi University, No. 300 Syuefu Rd., Chiayi City 60004, Taiwan; mutolisp@mail.ncyu.edu.tw
[2]  Experimental Forest / Department of Forestry, National Chung Hsing University, No. 145 Xingda Rd., Taichung City 40227, Taiwan
*   Correspondence: cachiu@nchu.edu.tw; Tel.: +886-4-2284-0397

**Abstract:** *Trochodendron aralioides* Siebold & Zuccarini (Trochodendraceae) is a famous relic tree species. Understanding the comprehensive spatial distribution and likely impacts of climate change on *T. aralioides* in its main habitat—Taiwan—is of great importance. We collected occurrence data and bioclimatic data to predict the current and future (year 2050) distribution by ensemble distribution modeling on the BIOMOD2 platform. Visualization of occurrence point data revealed that the main population of *T. aralioides* was concentrated at medium altitudes and extended to both ends of Taiwan, being especially rich in the northern low mountains. A similar distribution pattern of occurrence probability was shown by ensemble prediction of the true skill statistic >0.8 models. Comparing the current and future distribution of *T. aralioides*, the overlay analysis with profile display demonstrated spatial turnover that revealed a discrepancy between different latitudes and altitudes. In the future climate, *T. aralioides* at the middle altitudes of central Taiwan could migrate upward, but its population in northern Taiwan could lose most of its habitat. Consequently, *T. aralioides* in the low mountains of northern Taiwan could be particularly in need of further conservation research, which is urgently required to mitigate climate change impacts.

**Keywords:** relic species; species distribution modeling; BIOMOD2; spatial turnover; Yangmingshan National Park

## 1. Introduction

*Trochodendron aralioides* Siebold & Zuccarini (Trochodendraceae) is a famous broad-leaved tree and one of the most widespread cultured species in the world's botanical gardens. Geologically, the genus *Trochodendron* once occurred in western North America, Japan, and Kamchatka [1]. Currently, only the relic species *T. aralioides*, restricted to East Asia, remains [2,3]. *T. aralioides* is scattered across Taiwan but rare in Japan [4,5]. To date, we know that *T. aralioides* is common in the humid mid-altitude forests of central Taiwan [6–8] and lowlands of northern Taiwan [2,9]; however, its comprehensive distribution has remained uninvestigated. Furthermore, *T. aralioides* has been generally seen as an indicator of the cloud forest in Taiwan [6], and thus it is likely somewhat susceptible to climate change [10].

Modern species distribution modeling (SDM) began with the creation of the BIOCLIM package in 1984 [11,12]. This provided the bioclimatic variables and climatic interpolation methods that still underpin many studies. SDM is based on the equilibrium assumption that the species is in equilibrium with its environment [13]. Regardless of the limitations of the equilibrium assumption [14,15], SDM has been widely applied in various biological and environmental issues [16]. In recent years, interest

in and related articles on SDM have increased markedly [17]. Searching the "species distribution model" topic in the Web of Science Core Collection database (http://apps.webofknowledge.com/) revealed seven results during 1990–1999, 123 during 2000–2009, and as many as 1807 from 2010 to 2018. SDM is a powerful tool to support biogeography, biological conservation, and climate change [18–20]. It has offered new possibilities to explore biological distribution not only in the current environment but also in future climate change scenarios [21–23]. Expanded applications of SDM have benefited from rapid developments in modeling methods (e.g., maximum entropy modeling [24]; https://biodiversityinformatics.amnh.org/open_source/maxent/), species occurrence information (e.g., global biodiversity information facility (GBIF); http://www.gbif.org/), and environmental layers (e.g., WorldClim; http://www.worldclim.org/) available as free and open-source software or data.

The genus *Trochodendron* has only one relic species remaining, namely *T. aralioides*. Taiwan is its main habitat, and the *T. aralioides* population might be sensitive to climate change. Thus, it was necessary to understand the comprehensive spatial distribution of and climate change effects on the relic *T. aralioides* in Taiwan. This paper examined *T. aralioides* to (1) collect species occurrence data, (2) use SDM to model the spatial distribution, and (3) assess likely climate change impacts.

## 2. Materials and Methods

### 2.1. Study Area

Taiwan is a subtropical island in eastern Asia situated at the junction of the Ryukyu Island Arc and Philippine Island Arc (Figure 1). Its total area is ca. 36,000 km$^2$, with altitudes ranging from 0–3952 m above sea level. The climate is dominated by the East Asian monsoon, with the northeast monsoon during the winter half-year and southwest monsoon during the summer half-year. The annual mean temperature ranges from 4.0 to 25.0 °C from alpine to southern lowland; the annual precipitation is 1023–4880 mm [25]. Alternating winter and summer monsoons plus sharp and complex topography are the basic environmental conditions in Taiwan [6], creating a variety of plant habitats. Moreover, Taiwan provided refuge for many species during glacial periods when a large land bridge extended from eastern China to Taiwan and the Ryukyus, and likely to the main islands of Japan [26]. Owing to location, climate, topography, and glacial oscillation, more than 4000 taxa of vascular plants have been recorded by *Flora of Taiwan* [27], including our target species, *T. aralioides*—one of the best known broad-leaved relic tree species.

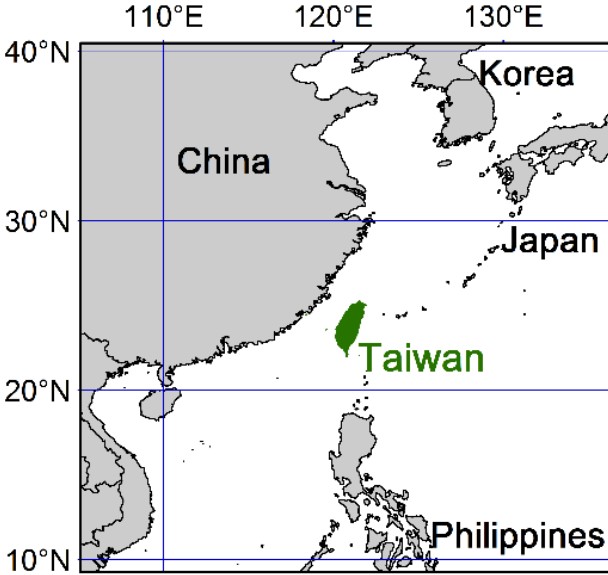

**Figure 1.** Location of the study area.

*2.2. SDM*

We used SDM to explore the spatial patterns of *T. aralioides* in current and future environments. SDM is composed of three main aspects: species occurrence data (dependent variable), environmental variable layers (independent variables), and a modeling algorithm [13,28,29].

Species occurrence data are the coordinate records of where individuals of *T. aralioides* were collected from biological databases and observed in the field by using the global positioning system (GPS). Occurrence data were collected from two databases: GBIF and the Taiwan Vegetation Diversity Inventory and Mapping Project (TVDIM) [30].

Many researchers have suggested that more attention should be given to the explanatory power and ecological basis for choosing environmental variables [28,31,32], especially those that are physiologically relevant [33]. However, the availability of some physiologically relevant variables was low [34], and thus we used bioclimatic variables as surrogates [35,36]. This study extracted 19 bioclimatic variable layers from the WorldClim database (http://www.worldclim.org) for current and future (2050) environments. These 19 bioclimatic variables (BIO1–BIO19) were derived from monthly temperature and rainfall values to generate more biologically meaningful variables [37]. The future climate scenario for the year 2050 (average for 2041–2060) used the same 19 variables downloaded from the WorldClim database projected by CCSM4 under the RCP4.5 scenario. Figure 2 partially demonstrates the current and future climate layers used as independent variables in SDM.

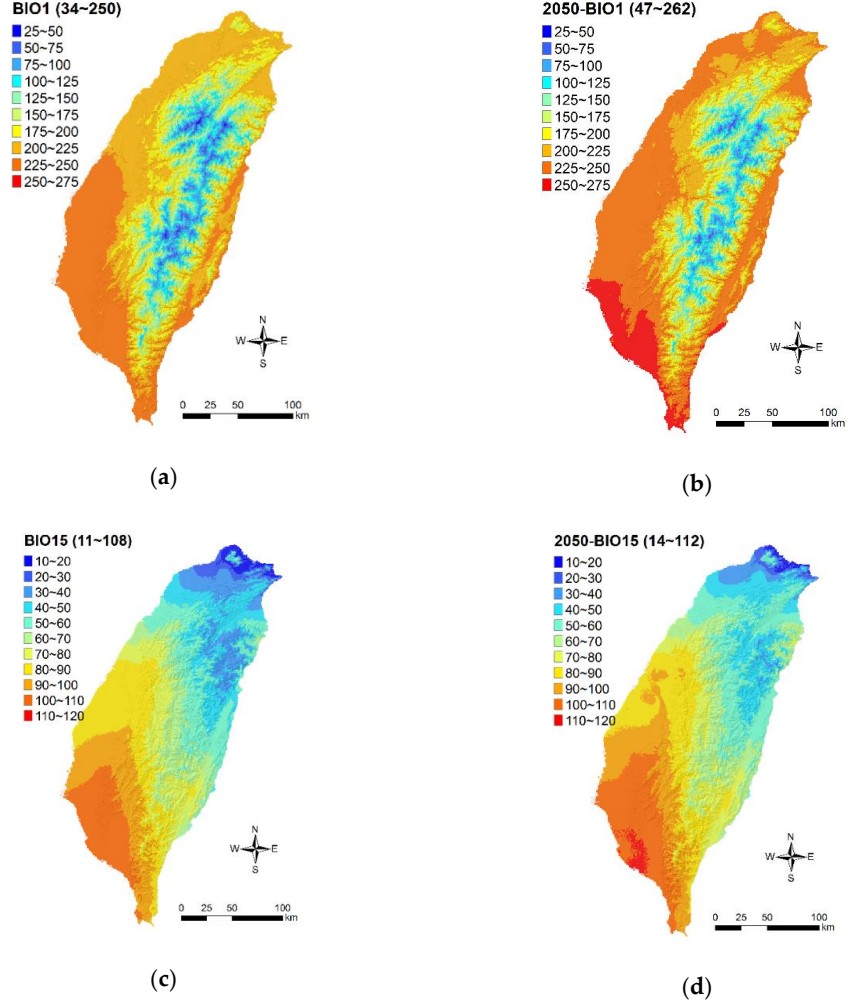

**Figure 2.** Current (**a**,**c**) and future (2050; **b**,**d**) bioclimatic variable layers: BIO1, annual mean temperature; BIO15, precipitation seasonality.

Regardless of the comparison and selection among different SDM methods, we used as many methods as possible and then chose those with high accuracy. Ten SDM methods were performed on the latest version of the BIOMOD2 package within the statistical software R (http://cran.r-project.org/web/packages/biomod2/index.html) [38], including Artificial Neural Networks (ANN), Classification Tree Analysis (CTA), Flexible Discriminant Analysis (FDA), Generalized Additive Models (GAM), Generalized Boosting Model (GBM), Generalized Linear Models (GLM), Breiman and Cutler's Random Forest for classification and regression (RF), Multiple Adaptive Regression Splines (MARS), MAXimum ENTropy modeling (MAXENT), and Surface Range Envelope (SRE). All models were run using the default options of the BIOMOD2 package and calibrated (85% occurrence dataset) and evaluated (15% occurrence dataset) using 25-fold cross-validation. There were 250 models from 10 techniques × 25 folds.

### 2.3. Accuracy Evaluation and Ensemble Forecasting

The performance of 250 models was evaluated using the true skill statistic (TSS) [39]—a threshold-dependent evaluation (sensitivity + specificity − 1). Liu et al. [40] suggested maximizing the sum of sensitivity and specificity for the threshold. Ben Rais Lasram et al. [41] used the following accuracy classification scheme: excellent (TSS > 0.8), good (TSS = 0.6–0.8), fair (TSS = 0.4–0.6), poor (TSS = 0.2–0.4), and no predictive ability (TSS < 0.2).

The BIOMOD2 package provided ensemble forecasting [38,42,43] that was relatively robust against uncertainty in the individual model [44] and was used to assess the effects of climate change [45]. In the present study, the predictive distribution maps of *T. aralioides* for both current and future climates resulted from ensemble forecasting in the BIOMOD2 package by parameterizing TSS > 0.8 for 250 individual models.

## 3. Results and Discussion

### 3.1. Occurrence Data of T. aralioides

A total of 1051 coordinated records of *T. aralioides* were extracted from the GBIF and TVDIM databases. Error and bias occurrence data such as those from the ocean, Lat/Long error, and Lat/Long zero were filtered out of the records [46]. Additionally, we used the GPS to obtain 1981 coordinated records of native *T. aralioides* individuals throughout Taiwan. All 3,032 coordinated records (Figure 3) were used as occurrence point data of *T. aralioides* in subsequent SDM. Figure 3 shows that the middle-altitude mountainous areas of the Central Mountain Range were the major distribution areas of *T. aralioides* [2,9], and the spatial pattern of *T. aralioides* distribution descended to Taiwan's northern and southern ends. The change in elevation was particularly evident in northern Taiwan [6] owing to the effects of a latitudinal increase and the northeast monsoon during the winter half-year [47]. The quality and quantity of species occurrence data—a total of 3032 presence points—facilitated the accuracy of SDM [16,18,48].

### 3.2. Ensemble Distribution Modeling

We used the ensemble SDM platform and R package BIOMOD2 [43] to explore the comprehensive spatial pattern of *T. aralioides*. Figure 4 shows box plots for TSS scores of the 250 models. The most accurate technique was RF and the least accurate was SRE. Although this finding was consistent with those of most studies (e.g., [49]), it differed from those of a few studies (e.g., [50]). An ensemble prediction combining all excellent models may be safer and could overcome uncertainty in model selection [51,52]. Among the 250 models, 210 had TSS > 0.8 or excellent accuracy based on the classification of Ben Rais Lasram et al. [41]. These high-accuracy predictive distribution maps were combined to form ensemble forecasting of *T. aralioides*, as shown in Figure 5a.

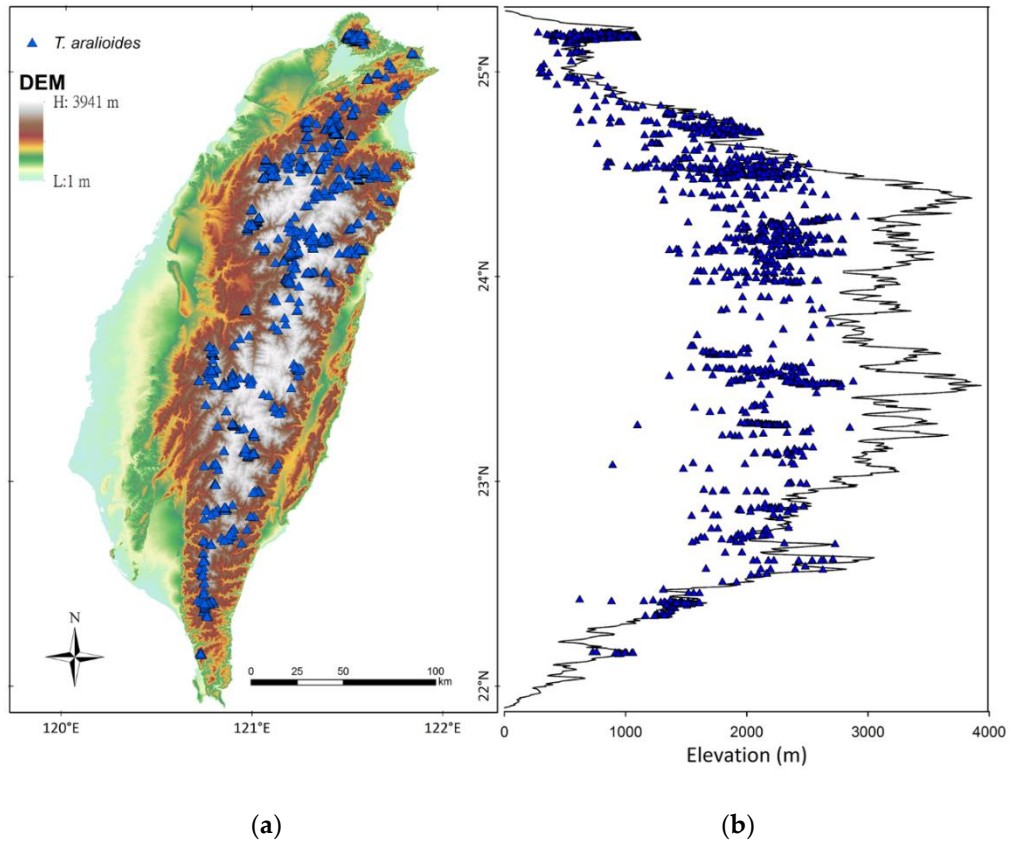

(**a**)                    (**b**)

**Figure 3.** A total of 3032 occurrence points of *T. aralioides* in Taiwan: (**a**) top view and (**b**) side view. DEM: Digital Elevation Model.

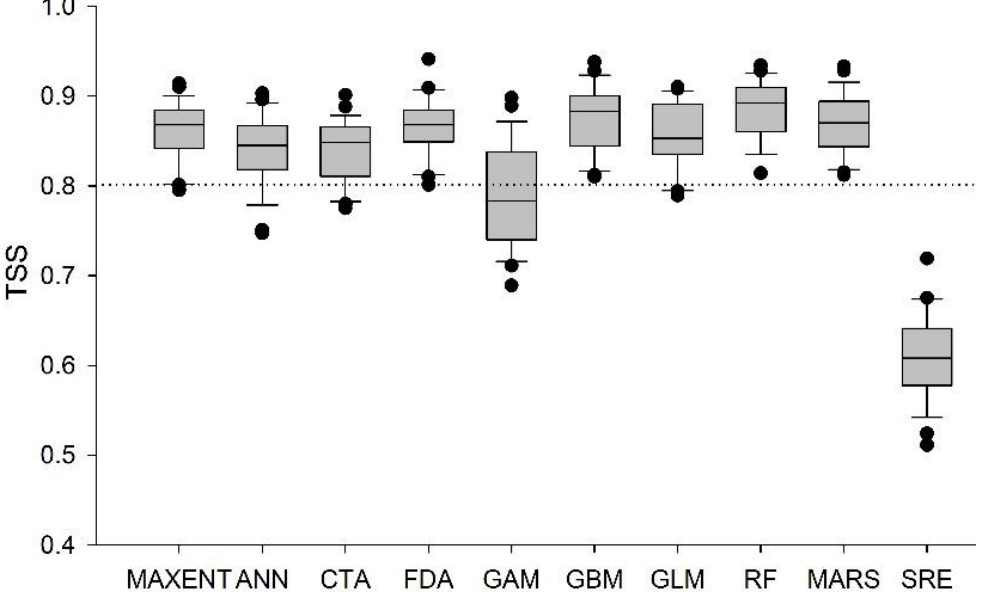

**Figure 4.** True skill statistic (TSS) of ten modeling methods for predicting *T. aralioides* distribution. MAXENT: Multiple Adaptive Regression Splines (MARS), MAXimum ENTropy modeling; ANN: Artificial Neural Networks; CTA: Classification Tree Analysis; FDA: Flexible Discriminant Analysis; GAM: Generalized Additive Models; GBM: Generalized Boosting Model; GLM: Generalized Linear Models; RF: Breiman and Cutler's Random Forest for classification and regression; MARS: Multiple Adaptive Regression Splines; SRE: Surface Range Envelope.

On the whole, the humid mid-altitude forest of central Taiwan was the principal distribution area of *T. aralioides*, consistent with the description of *Flora of Taiwan* [7]. The majority of the population was concentrated in the forest edges of the *Quercus* zone, *Chamaecyparis* forest, and mixed coniferous and evergreen broad-leaved forest [6,9,53]. However, *T. aralioides* did not form a pure stand but was scattered and mixed with many coniferous and broad-leaved species. The broad-leaved *T. aralioides* and coniferous *Chamaecyparis formosensis* Matsum. are the most representative indicator plants of the cloud forest zone in subtropical Taiwan [6]. Moreover, Figure 5a reveals that another hotspot occurred in the lowlands of northern Taiwan [2,9] or the low-montane warm-temperate rainforest [54]. The lowland position facing the cold and damp northeast monsoon as well as increasing latitude was probably the main cause of downward migration of *T. aralioides*.

### 3.3. Climate Change Effects

To assess the climate change effects on *T. aralioides*, we projected 10 SDM models to the future (2050) climate scenario. As with the above-mentioned processes, the high-accuracy predictive distribution maps (TSS > 0.8) from 250 individual SDM models were used to integrate predictive results. Figure 5b demonstrates the ensemble forecasting of *T. aralioides* distribution under future climate scenarios. Visually, its overall future distribution seems to be in decline.

To further examine the spatial change of *T. aralioides* distribution in current and future climate scenarios (Figure 5a vs. Figure 5b; Table 1), overlay analysis was performed (Figure 6a). The difference in occurrence probability of *T. aralioides* between current and future climates was between −0.657 and 0.575 and could be divided into 11 parts. As shown in Figure 6a, the gray area represents almost no difference in *T. aralioides* distribution; the warm colors represent the gain area, mainly in the mid-high mountains of central Taiwan; and the cool colors represent the loss area, mainly in the mid-low mountains of northern Taiwan. To calculate the overall distribution, the occurrence probability was reduced by 0.023 in the future climate and in line with the aforementioned visual observations.

Figure 6b is the side view of Figure 6a with the same color system. Each circle in Figure 6b corresponds to one grid in Figure 6a, and the jagged line in the Figure 6b background is the peak profile of Figure 6a. Figure 6b presents the change in *T. aralioides* spatial patterns. The warm colors, namely the gain area, in the central mountains indicate upward migration of *T. aralioides* to nearly ca. 3500 m above sea level. In contrast to Figure 3b, the upper limit of the current *T. aralioides* distribution was ca. 3000 m above sea level. Temperature rise was considered a critical factor driving mountainous plants to migrate upward [55–57].

**Table 1.** Average occurrence probability and change of *T. aralioides* predicted in current and future climates in different elevation zones.

| Elevation Zone (m) | Average Occurrence Probability | | |
| --- | --- | --- | --- |
| | **Current** | **Future** | **Future–Current** |
| 0–499 | 0.027 | 0.021 | −0.006 |
| 500–999 | 0.169 | 0.096 | −0.073 |
| 1000–1499 | 0.370 | 0.260 | −0.110 |
| 1500–1999 | 0.693 | 0.607 | −0.087 |
| 2000–2499 | 0.811 | 0.830 | 0.019 |
| 2500–2999 | 0.532 | 0.727 | 0.195 |
| 3000–3499 | 0.181 | 0.368 | 0.187 |
| 3500–3999 | 0.134 | 0.172 | 0.038 |

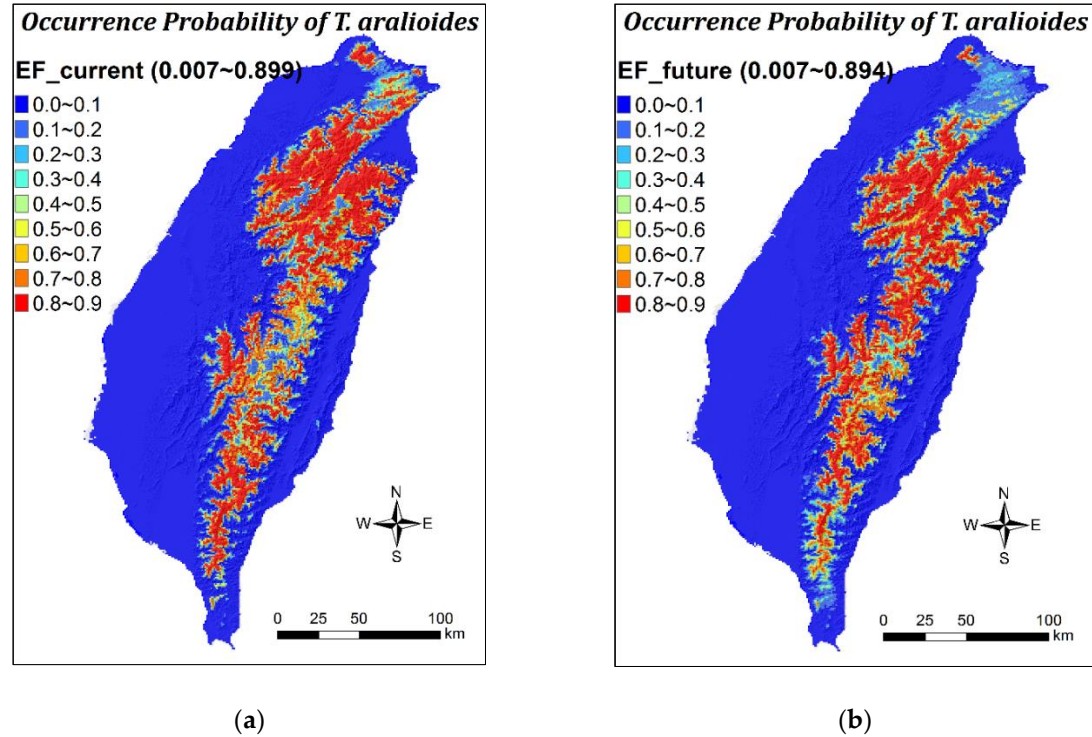

**Figure 5.** Ensemble forecasting of *T. aralioides* distribution under (**a**) current climate and (**b**) future climate (year 2050, projected by CCSM4 under the RCP4.5 scenario).

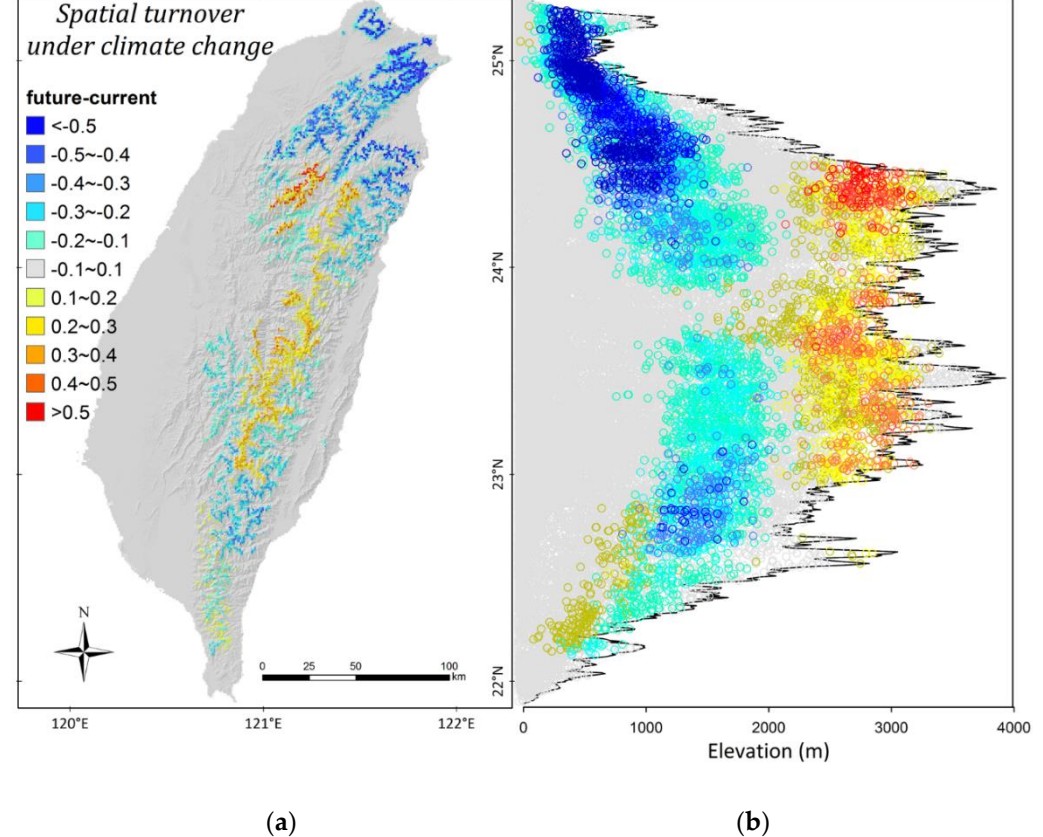

**Figure 6.** Overlay analysis of *T. aralioides* spatial distribution in current and future climate scenarios: (**a**) top view and (**b**) side view.

The loss area in northern Taiwan was considerably greater (Figure 6b), indicating a significant reduction in the distribution area. The most severe decline occurred in Yangmingshan National Park and its adjacent mountains in northern Taiwan (Figure 6). *T. aralioides* naturally grew well and was dominant in Yangmingshan, which contains the only pure forest of *T. aralioides* in the world (Figure 7a). The *T. aralioides* populations in Yangmingshan exhibited lower genetic variation than their central Taiwanese counterparts [58]. The altitudes of the Yangmingshan populations were much lower than those of the central populations [8].

The natural seedling establishment of *T. aralioides* was most frequently found on moist mossy ground (Figure 7b). The rich population of *T. aralioides* in northern Taiwan was a result of the combined effects of lower temperature from the higher latitude and northeast monsoon [47] and a more humid environment [59]; however, under the future warming climate scenario the population could lose its habitat range (Figures 5 and 6) because the trees are already situated on mountaintops, and thus cannot migrate further upward [60,61]. Consequently, we must pay special attention to the population of *T. aralioides* in northern Taiwan based on its sensitivity to climate change and gene specificity.

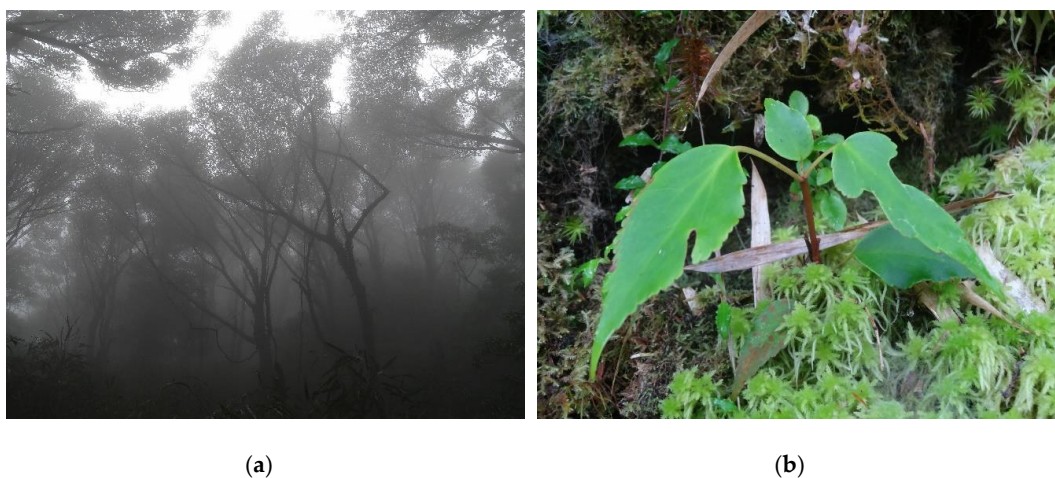

(**a**)　　　　　　　　　　　　　　　　　　(**b**)

**Figure 7.** (**a**) The only pure forest of *T. aralioides*, located in Yangmingshan. (**b**) Natural seedling establishment of *T. aralioides* on moist mossy ground.

## 4. Conclusion

Taiwan is the world's primary habitat for the relic species *T. aralioides*. Both the occurrence point data and ensemble distribution modeling by BIOMOD2 in this study revealed that the two largest populations of *T. aralioides* appeared in the middle altitudes of central Taiwan and lower mountains of northern Taiwan. Overall, the species' habitat could gradually become compressed under future (year 2050) climate conditions, primarily because the temperature is projected to be 1.2–1.4 °C higher than that of today, as estimated by the Intergovernmental Panel on Climate Change's AR5 RCP4.5 scenario. Such a warming climate could cause the middle-altitude *T. aralioides* in central Taiwan to migrate upward, whereas the *T. aralioides* in northern Taiwan could lose most of its habitat. Therefore, this area is critical for further conservation research.

**Author Contributions:** Both authors designed this research, analyzed the results, and composed the manuscript.

**Funding:** This research was funded by Yangmingshan National Park Headquarters grant number PG10301-0407.

**Conflicts of Interest:** The authors declare no conflict of interest.

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
