# Peer review of "The Relic Trochodendron aralioides Siebold & Zucc. (Trochodendraceae) in Taiwan: Ensemble Distribution Modeling and Climate Change Impacts"

_forests, doi:10.3390/f10010007_

Round 1
Reviewer 1 Report
Dear Authors,
Your manuscript is well written and the aims, methods and results are presented in a clear way. However, I feel that some more information should be given concerning the models. Why did you select these models and how do they differ? It would also be good if you could visualize or describe the model uncertainty more, e.g. by presenting the residuals or CV errors.
Last, but not least, I suggest that you have the same color scale for (a) and (b) as well as (c) and (d) in Figure 2 and (a) and (b) in Figure 5 to make the visual comparisons easier.
Best regards
Author Response
Please refer to the attached PDF file.

Reviewer 2 Report
This paper applies species distribution modelling (SDM) to examine current and likely future distributions of a relic tree species. It is generally a well prepared and presented analysis that should be of interest to Forests readers. I recommend that it should be accepted following minor modifications. I attach an edited pdf and some additional comments are provided below.
Title – Appropriate – possibly change “climate change effects” to “change change impacts”
Abstract – Adequate – a few minor changes have been suggested in the edited pdf.
Introduction – Generally adequate, but there is a need to acknowledge the pioneering SDM work and mention the ‘equilibrium assumption’.
L.40 See suggested added couple of sentences in edited pdf. To acknowledge the original SDM work you should cite the following two references:
Nix, H.A. (1986) A biogeographic analysis of Australian elapid snakes. Atlas of elapid snakes of Australia: Australian flora and fauns series 7 (ed. R. Longmore), pp. 4-15. Bureau of Flora and Fauna, Canberra.
Booth T. H., Nix H. A., Busby J. R. & Hutchinson M. F. (2014) BIOCLIM: The first species distribution modelling package, its early applications and relevance to most current MAXENT studies. Divers. Distrib. 20, 1–9.
You’ll see in the 2014 paper that the first SDM climate change studies were published in 1988. One involved a native forest and another a plantation. The set of 19 bioclimatic variables used in your paper was developed for BIOCLIM in 1996 and WORLDCLIM was created using climatic interpolation routines developed originally for BIOCLIM. This would all be covered by citing the 2014 reference.
Your study involves what is known as the ‘equilibrium assumption’. See following extract from 2018 paper on “Why understanding the pioneering and continuing contributions of BIOCLIM to species distribution modelling is important”. This paper mentions the first two SDM climate change studies:
“Busby (1988) stated what has become known as the ‘equilibrium assumption’ at the start of his discussion noting that “the primary assumption of the BIOCLIM system is that entities can only colonise and survive in areas with climates fitting within their present climate profile”. Ironically, the Booth and McMurtrie (1988) paper showed how widely P. radiata is grown in Australia,
and thus provided some evidence why this assumption should be viewed with caution for long-lived
tree species under climate change. P. radiata is an endangered species in its natural distribution of just 5300 ha in California, but is the most widely planted softwood species in the southern hemisphere with over 4 M ha of plantations demonstrating its considerable climatic adaptability (Mead 2013).”
It’s reasonable to assume that under climate change a long-lived tree may be able to display some of the climatic adaptability that it shows in trials outside its natural distribution. You do not need to cite the 1988 studies, but you should at least mention the equilibrium assumption in the introduction. There’s a 2017 paper on “Assessing species climatic requirements beyond the realized niche (Climatic Change, 145, 259-271) if you want to know more. Looking at the GBIF map (https://www.gbif.org/species/4191567) it would appear that there may be little evidence of T. aralioides being suitable for warmer climates than Taiwan, so if that is so you could simply note it in the introduction.
L. 41 I only get 1037 hits from the Web of Science core collection when searching for “species distribution model” for any year – please check that your 4,457 hits for 2010 to 2018 is correct.
Materials and Methods – Adequate, just a couple of minor changes suggested in the edited pdf.
Results and Discussion
L. 127 Occurrence data – it’s good to see such a large collection of data points.
L. 132 It’s not surprising that SRE (a BIOCLIM-type approach) performed relatively poorly. It may be worth noting that the use of 35 variables has been recommended for BIOCLIM since 1999. SRE approaches will always perform poorly when used with relatively few (e.g. 19) variables. When used with 35 variables BIOCLIM performs almost as well as Maxent for both current and climate change predictions, but here you only had ready access to the 19 bioclimatic variables from WORLDCLIM. See the Penman et al. (2010) paper (Diversity and Distributions 16, 109-118) cited in your reference 14 for a 35 variable comparison of BIOCLIM and Maxent. I use and recommend Maxent, but BIOCLIM is not as bad as it is often portrayed if it is used correctly.
Figure 3 – It’s a good idea showing the side (i.e. elevation) as well as the conventional view.
Figure 5 – It would be good to indicate the year of the future forecast i.e. 2050 and the scenario used
i.e. CCSM4 under the RCP4.5 scenario in the caption.
Figure 6 – The differences between Figures 5 a and b aren’t immediately obvious, so Figure 6 is very useful. Again, the side view is very helpful.
Conclusion – adequate.

Author Response
Please refer to the attached PDF file.
